# Examining Language, Speech and Behaviour Characteristics: A Cross-Sectional Study in Saudi Arabia Using the Arabic Version of Gilliam Autism Rating Scale-Third Edition

**DOI:** 10.3390/children11040472

**Published:** 2024-04-15

**Authors:** Muhammad Alasmari, Ahmed Alduais, Fawaz Qasem, Shrouq Almaghlouth, Lujain AlAmri

**Affiliations:** 1Department of English Language and Literature, College of Arts of Letters, University of Bisha, Bisha 67714, Saudi Arabia; moaalasmri@ub.edu.sa (M.A.); faqasem@ub.edu.sa (F.Q.); 2Department of Human Sciences (Psychology), University of Verona, 37129 Verona, Italy; 3Department of English Language and Translation, College of Science and Theoretical Studies, Saudi Electronic University, Riyadh 11673, Saudi Arabia; sa.almaghlouth@seu.edu.sa; 4Speech Pathology Division, Jeddah Institute for Speech and Hearing and Medical Rehabilitation, Jeddah 21471, Saudi Arabia; lujain@jish.med.sa

**Keywords:** Gilliam Autism Rating Scale—Third Edition, Arabic GARS-3, autism spectrum disorder, developmental status, Saudi Arabia

## Abstract

Autism spectrum disorder (ASD) exhibits diverse manifestations influenced by demographic factors. This study evaluates these variations within Saudi Arabia, aiming to investigate language, speech and behaviour characteristics across different demographics in Saudi Arabia using the Arabic Version of the Gilliam Autism Rating Scale—Third Edition (A-GARS-3). Employing a cross-sectional design, 178 participants were stratified by developmental status (n = 124 school settings, n = 54 clinical setting), sex (Females = 77, Males =101), age (range = 3–22), and geographical region (different provinces in Saudi Arabia). The A-GARS-3 measured ASD manifestations across six subscales. The study identified significant differences in ASD manifestations by developmental status, with higher ASD likelihood and severity in clinical settings. Younger children showed more pronounced ASD characteristics, and males were slightly more likely to be diagnosed with ASD. Geographical analysis revealed regional differences in severity. The findings underline the importance of demographic considerations in ASD assessment and diagnosis, suggesting the need for age-specific and culturally sensitive approaches. The A-GARS-3 is a reliable tool for the Saudi context. Regional disparities in ASD prevalence and severity indicate a need for tailored health policies and resources across Saudi provinces.

## 1. Introduction

### 1.1. An Overview of Autism Spectrum Disorder

Autism spectrum disorder (ASD) is a complex developmental condition marked by substantial obstacles in social interaction and communication, coupled with a repertoire of repetitive and restricted behaviors. The World Health Organization’s International Classification of Diseases, Eleventh Revision (ICD-11), delineates ASD as encompassing enduring difficulties in social reciprocity and communicative skills, as well as rigid behavioral patterns that deviate from the normative expectations of the individual’s age and cultural background [1]. Manifesting primarily during the developmental phase, typically in early childhood, ASD can engender functional impairments across several domains. Parallel to the ICD-11, the American Psychiatric Association’s Diagnostic and Statistical Manual of Mental Disorders: DSM-5-TR, enumerates similar diagnostic criteria, including persistent challenges in social communication and interaction across diverse settings, along with restricted and repetitive behavior patterns [2]. Both diagnostic manuals acknowledge the spectrum nature of ASD, with the DSM-5-TR noting the necessity for symptoms to emerge in the early developmental stages and the ICD-11 highlighting gender disparities in diagnosis rates, with a higher prevalence in males. These diagnostic standards are instrumental in the prompt identification of ASD, which is crucial for initiating early interventions and providing support for the affected individuals and their families [1,2].

### 1.2. Impact on Families and Heterogeneity of ASD

Further, the impact of an ASD diagnosis on family quality of life is profound and multifaceted. Families navigating the early stages of an ASD diagnosis often face considerable stress, which can strain family dynamics and emotional well-being. The study highlights how parental stress, particularly in relation to the core symptoms of ASD, negatively correlates with the family’s quality of life. This stress can manifest in various ways, affecting the mental health of family members, their relationships, and even their financial stability, as they may need to address the specialized needs of the individual with ASD, including therapy, education, and medical care [3].

The recognition of ASD’s heterogeneity has been widely regarded as an accurate reflection of the disorder’s diverse manifestations [4]. This diversity underscores the fundamental principle that a plethora of internal and external variables interact to create the distinct profiles observed within the ASD spectrum. Such complexity has not inhibited substantial advancements in this field. On the contrary, the critical need for increased awareness, comprehensive screening, and early intervention is a recurrent theme in the literature, driving significant progress in the field [5,6,7,8,9]. Research has consistently demonstrated that early detection can profoundly reduce distress, preserve cognitive function, and enhance the quality of life and independence for individuals with ASD who engage in intervention programs. Furthermore, an ASD diagnosis has a deep and multifaceted impact on family quality of life. Families confronting an ASD diagnosis often encounter significant stress, which can disrupt family dynamics and emotional health. A recent study by Papadopoulos et al. highlights the negative correlation between parental stress, especially in relation to ASD’s core symptoms, and the family’s quality of life [3]. This stress can affect various aspects, including the mental health of family members, their interpersonal relationships, and economic security, as they may incur additional expenses for specialized care, including therapy, educational support, and medical treatments.

Despite ASD’s acknowledged heterogeneity, a central theme persists across the spectrum: deficits in social interaction and communication, along with a limited propensity to engage in play or activities [10]. The manifestations of these core features of ASD are highly variable, reflecting the diverse factors at play and the disorder’s presentation at different life stages. For analytical purposes, we may discuss these characteristics separately; however, they typically intermingle in real-life scenarios without clear demarcation. The Centers for Disease Control and Prevention (CDC) notes that ASD can be identified in children as young as 12 to 14 months, with early signs including avoidance of eye contact, lack of response to their names by 9 months, disinterest in interactive games by 12 months, and absence of pointing to objects of interest by 18 months [11]. By 24 months, children with ASD may not recognize others’ emotions, by 36 months they may not play with peers, and by 60 months may show no interest in role-playing or imaginative activities. Repetitive behaviors, such as echolalia, insistence on sameness, hand-flapping, or spinning, are also common. Often, children with ASD exhibit delays in language, motor skills, or learning, and may experience anxiety, and unusual sleep and eating habits [11].

The majority of ASD research has traditionally focused on young children and the early developmental stages, with a particular emphasis on the importance of early detection and intervention for improving life outcomes for those diagnosed [12]. Nevertheless, Schall and Todd McDonough point out the relative scarcity of research on adolescents and adults with ASD, which is noteworthy given that ASD is recognized as a lifelong condition [13]. They suggest that the wide range of heterogeneity observed in children with ASD further diversifies in later stages, potentially due to access to adequate professional and social support. Furthermore, they note that while autistic traits are often more pronounced at a younger age, ASD is a neurodevelopmental condition distinguished by significant challenges in social interaction, communication skills, and a tendency for restricted and repetitive behaviors [13]. The World Health Organization, through its International Classification of Diseases, Eleventh Revision (ICD-11), characterizes ASD by enduring impairments in social communication and interaction, coupled with rigid behavior patterns that are incongruent with the individual’s developmental stage and cultural context [1]. Emerging typically during early childhood, ASD can lead to considerable dysfunction across multiple life areas. The Diagnostic and Statistical Manual of Mental Disorders, Fifth Edition, Text Revision (DSM-5-TR) by the American Psychiatric Association provides parallel diagnostic criteria, specifying that these impairments must be present across various contexts and feature prominently in the individual’s early developmental phase [2]. Both the ICD-11 and the DSM-5-TR acknowledge that ASD manifests with considerable variability, with the DSM-5-TR particularly noting the requirement for early developmental onset of symptoms and the ICD-11 acknowledging differences in cultural presentation and a higher diagnostic prevalence in males. The establishment of these diagnostic frameworks plays a pivotal role in the early detection of ASD, which is crucial for timely intervention and support for affected individuals and their families [1,2].

### 1.3. Factors Influencing the Presentation and Management of ASD

The presentation and management of ASD are influenced by several key factors, including age, sex, developmental status, and geographical location. Early detection of ASD, critical for effective intervention, can manifest through developmental markers as early as 12 to 14 months, although symptom presentation and needs evolve throughout the lifespan [11,13]. Sex differences are evident, with higher rates of diagnosis in males, potentially due to both biological differences and diagnostic biases, whereas females may be underdiagnosed due to subtler symptoms [14]. Co-occurring psychiatric conditions can complicate the ASD profile, influencing both diagnosis and intervention approaches [15]. Additionally, geographical disparities impact the prevalence, recognition, and resources available for ASD, with variations in cultural perceptions and healthcare infrastructure affecting how ASD is managed across different regions [16]. These factors highlight the necessity for a personalized approach in the treatment and support of individuals with ASD.

### 1.4. Purpose of the Present Study

The aim of this study is to conduct a targeted evaluation of ASD manifestations within a Saudi Arabian context, focusing on a modest cohort of 178 participants to identify potential demographic variations. The objectives include: (1) assessing the relationship between developmental status, sex, age groups, and geographic locations within Saudi Arabia and ASD symptomatology; and (2) determining the prevalence and severity of ASD symptoms within these demographic subgroups using the Arabic Version of the A-GARS-3. By adopting a cross-sectional study design, this research intends to provide a detailed understanding of how ASD presents in a Saudi Arabian context. The study’s refined scope, considering the limited sample size, will contribute to the larger discourse on culturally responsive diagnostic practices and inform more targeted health policies and resource allocation within Saudi Arabia’s diverse communities.

## 2. Method

### 2.1. Sample

In this cross-sectional study assessing the utility of an Arabic adaptation of the A-GARS-3 within Saudi Arabia, the demographic distribution of participants highlighted a dichotomous sample composition across educational and clinical settings (See Table 1). Of the 178 randomly sampled individuals, 124 were ascertained from school environments, encapsulating 69.66% of the total, while the remaining 30.34% (*n* = 54) were derived from clinical contexts. The age demographics indicated a preponderance of younger participants in the clinical setting, particularly within the ‘3–5’ age bracket, which constituted 51.85% as compared to 22.58% in the school setting. A stark gender disparity was observed, with the clinical cohort comprising a male majority at 68.52%. Geographical representation was notably disproportionate, with the Asir province predominantly represented in schools (72.58%), and Mecca province preponderant in clinical settings (92.59%). Exceptionality status was exclusively reported in clinical settings, with speech and language delay being the most prevalent at 50%. The participant pool was homogeneously Arab, speaking the Saudi dialect of Arabic, which ensures cultural and linguistic consistency with the adapted assessment tool.

### 2.2. Instrument: The Translation Process

The GARS-3 is an established and widely recognized instrument designed for the evaluation and diagnostic assessment of ASD within individuals aged 3 to 22 years [17]. This tool is constructed to reflect the diagnostic criteria outlined in the DSM-5, and it incorporates a tripartite scale system—consisting of subscales for social interaction, social communication, and restricted/repetitive behaviours—thus encapsulating the core characteristics associated with ASD. In the English version of GARS-3, the consistency and reliability of these subscales have undergone rigorous empirical validation, as seen in the scholarly critique and subsequent affirmative recognition of their internal consistency and inter-rater reliability. Such endorsement is mirrored in studies that have critically examined the construct and diagnostic validity of the GARS-3, with research underscoring its utility in both clinical and research populations, revealing the scale’s methodological robustness and its nuanced diagnostic capacity within the realm of ASD. The reported reliability for the original scale is 0.94 for the Autism Index 4 and 0.93 for the Autism Index 6, and the range of Cronbach value for the subscales is between 0.79 and 90 [17].

Conversely, the A-GARS-3 has been meticulously tailored to address the linguistic and cultural nuances inherent within the Arab-speaking demographic. The adaptation process began with a direct translation, followed by a comprehensive review by a panel of experts in linguistics, translation, and speech–language pathology. This was succeeded by a back-translation protocol to ensure semantic equivalence with the English version. The A-GARS-3 was then subjected to a pilot study to evaluate its practical application and effectiveness. The translation retained the original instrument’s six subscales, which were directly transposed to assess ASD-related behaviours in a culturally congruent manner. The A-GARS-3’s translation integrity and cultural relevance were paramount, with careful considerations made regarding idiomatic expressions and societal norms characteristic of the Saudi Arabian context, ensuring the tool’s resonance with the target population [17].

The psychometric properties of the A-GARS-3 have been substantiated through a comprehensive cross-sectional study that confirms its reliability and validity for use in Arabic-speaking regions [18]. The instrument demonstrated a high internal consistency, with Cronbach’s alpha and McDonald’s omega coefficients exceeding the threshold for excellent reliability. The instrument’s construct validity was affirmed through exploratory and confirmatory factor analyses, indicating a coherent underlying structure consistent with the original GARS-3’s theoretical constructs. Predictive validity was evidenced by significant correlations between the Autism Index—a composite score derived from the subscale scores—and each subscale, especially those measuring social interaction and communication. The rigorous validation process of the A-GARS-3, including factorial and statistical analysis, ensures that the scale is a robust and reliable tool for diagnosing ASD within the cultural framework of Arab societies, promising enhanced screening, diagnosis, and subsequent intervention strategies tailored to this specific linguistic and cultural context. The scale exhibits excellent internal consistency, with Cronbach’s alpha and McDonald’s omega both indicating very high reliability (α = 0.971; ω = 0.972). The range Cronbach value for the six subscales was reported in our analysis between 0.77 and 0.88 [18].

The GARS-3 is structured with six subscales that intricately map onto the multifaceted nature of ASD, providing a comprehensive framework for assessment (See Table 2). The first subscale, assessing restricted and repetitive behaviours, captures the proclivity for uniformity and repetition that is often a hallmark of ASD. The second and third subscales delve into social interaction and social communication, respectively, gauging the individual’s ability to engage with others and the competence in using and interpreting both verbal and nonverbal forms of communication. Emotional responses, the fourth subscale, scrutinize the affective aspects, including the range and appropriateness of emotional reactions, which can be atypical in ASD. Cognitive style, the fifth subscale, evaluates distinctive patterns of thinking and processing, such as a preference for detail-focused and concrete thinking. Lastly, the maladaptive speech subscale inspects aspects of language use that are considered aberrant, such as echolalia or pronoun reversal. Each subscale consists of specific items designed to capture the essence of these domains, thereby enabling the GARS-3 to provide a detailed profile of autistic traits as they manifest in a given individual.

### 2.3. Design

The study employs a cross-sectional design to evaluate the presentation of ASD symptoms within the Saudi Arabian context, utilizing a specific sample of 178 participants to explore demographic differences. The research aims to: (1) examine the associations between developmental status, sex, age, and geographic location within Saudi Arabia and the expression of ASD symptoms; and (2) measure the prevalence and intensity of ASD symptoms across these demographic categories using the Arabic Version of the Autism Spectrum Rating Scales (A-GARS-3). This design provides a concise and current picture of ASD in Saudi Arabia, despite the relatively small sample size. The findings are expected to enhance understanding of ASD within this cultural setting and contribute to the development of culturally sensitive diagnostic procedures. The results will also assist in shaping health policies and directing resources effectively within Saudi Arabia’s varied communities.

### 2.4. Procedures

The integrity of the study’s procedures was rigorously maintained to ensure the reliability and validity of the data collection process. Throughout the research, informed consents were obtained from all participants within school and clinical settings to adhere to ethical standards and maintain the autonomy of the participants involved. In the educational context, both parents and teachers were provided with comprehensive training on how to use the Autism A-GARS-3. This extensive training included detailed explanations of each item on the scale, the importance of objective observation, and practical sessions to practice scoring. The training sought to equip them with the skills necessary to accurately observe and record the behaviours and characteristics of the participants consistent with the scale’s guidelines. This meticulous preparation was designed to ensure that when parents and teachers completed the A-GARS-3, their assessments would be carried out with a high degree of consistency and precision, accurately reflecting the observed behaviours of the children.

In contrast, within the clinical environment, the administration of the A-GARS-3 was entrusted to trained clinicians and speech–language pathologists who possessed specialized knowledge and experience in working with the clinical population. Their professional expertise allowed for a nuanced and sensitive approach to engaging with participants and carrying out the assessment, which was particularly important for accurately capturing the complex clinical profiles of individuals with ASD. The clinical evaluations were conducted with individuals who had previously been diagnosed with ASD, a fact that is documented in detail in Table 1 of the study. This dual approach, utilising both educational and clinical professionals, was designed to validate the use of A-GARS-3 across different settings and ensure that the tool was administered in a way that was both culturally sensitive and appropriate to the Saudi Arabian context.

The A-GARS-3 was operationalized via Google Forms, an approach that not only streamlined the collection process but also bolstered the integrity and confidentiality of the data. Participants’ responses were submitted online, directly integrating into the research team’s database, thereby minimizing the potential for data-entry errors and ensuring real-time capture of the responses. This digital transformation of the scale facilitated a seamless and efficient data aggregation process, integral to the large-scale deployment of the instrument.

Ethical considerations were paramount in the study’s design and execution. Prior to the commencement of the data collection, informed consent forms elucidated the research’s objectives, the voluntary nature of participation, and the measures in place to ensure anonymity and confidentiality. These forms were disseminated and retrieved, ensuring that participants and their guardians were fully informed and consented to the involvement in the study. The implementation of anonymity and confidentiality protocols served to protect participant identity and the sensitivity of the information provided, adhering to ethical research standards and fostering a foundation of trust between the researchers and participants. Above all, an institutional review board (IRB) approval was obtained from the Department of English Language and Literature, University of Bisha, Saudi Arabia.

The subsequent data analysis phase commenced with a rigorous data cleaning process, a critical step to ensure the validity and reliability of the findings. The data, initially collected in Arabic, were translated into English to accommodate the diverse linguistic capabilities of the research team and to ensure the data’s accessibility for analysis. Following translation, the data were transitioned from Excel to Jamovi (version 2.3.26.0), a statistical software that supports both descriptive and inferential statistical analyses. This transfer was conducted with precision, ensuring that data integrity was maintained across platforms. Within Jamovi, a range of statistical tests were performed, encompassing both descriptive statistics to capture the sample’s characteristics and inferential statistics to explore the relationships between variables and to test hypotheses. The application of these analyses provided a robust statistical foundation for the study’s conclusions, aligning with the overarching research objectives and advancing the field’s understanding of the scale’s utility within the target population.

## 3. Results

The initial phase of statistical examination involved creating survey plots for key variables such as developmental status, age group, and gender in relation to ASD. These plots took into account the likelihood of ASD, the nature of support needed, and the degree of ASD severity as classified by the Diagnostic and Statistical Manual of Mental Disorders, Fifth Edition (DSM-5) [19].

### 3.1. ASD and Developmental Status

Figure 1A–C shows the ASD and developmental status based on three factors: the probability of ASD in three levels, the possibility of required support using four descriptors, and the ASD severity level in three levels. It should be noted that these are based on the original GARS-2 version. Figure 1A suggests that ASD is more likely to be diagnosed in a clinical setting than in a school setting. It also suggests that ASD is more likely to be “very likely” in a clinical setting than in a school setting. Figure 1B suggests that students who are referred to a clinical setting are more likely to require substantial or very substantial support compared to students in a school setting. Figure 1C suggests that individuals diagnosed with ASD in a clinical setting are more likely to fall into the Level 3 (most severe) category compared to those diagnosed in a school setting. Conversely, individuals diagnosed in a school setting are more likely to fall into the “No ASD” category compared to those diagnosed in a clinical setting.

### 3.2. ASD and Age Groups

Figure 2A–C shows the ASD and different age groups based on three factors: probability of ASD in three levels, the possibility of required support using four descriptors, and the ASD severity level in three levels. Figure 2A seems to suggest that ASD is more likely to be diagnosed in adults. Figure 2B suggests that young children are less likely to require substantial or very substantial support in a school setting compared to children and adolescents and adults. Figure 2C suggests that ASD is more likely to be diagnosed in young children and adults.

### 3.3. ASD and Sex

Figure 3A–C shows the ASD and sex based on three factors: the probability of ASD in three levels, the possibility of required support using four descriptors, and the ASD severity level in three levels. Figure 3A suggests that ASD is more likely to be diagnosed in a clinical setting than in a school setting or the general population, for both females and males. It also suggests that ASD is slightly more likely to be diagnosed in males than in females, across all three settings. Figure 3B proposes that males with ASD are more likely to require very substantial support compared to females with ASD, regardless of the setting. Figure 3C the graph indicates that individuals with ASD diagnosed in a clinical setting are more likely to fall into the Level 3 (most severe) category compared to those diagnosed in a school setting. Conversely, individuals diagnosed in a school setting are more likely to fall into the “No ASD” category compared to those diagnosed in a clinical setting.

### 3.4. Descriptive Statistical Analysis of Age Groups and Settings

The second step was to run descriptive analysis, (See Table 3), which indicates distinct variation across age groups and settings in the appraisal of ASD-related characteristics. Within the cohort of young children, individuals in clinical settings displayed marginally higher mean scores in restricted/repetitive behaviours and social communication than their school counterparts, suggesting that clinical populations may exhibit more pronounced ASD symptomatology. This trend is also observed in the social interaction subscale, where clinical settings yield a higher mean, pointing towards greater social impairments within this environment. However, the cognitive style subscale presents an anomaly, wherein young children in school settings demonstrate higher mean scores, which may indicate a greater diversity in cognitive patterns among this population.

Children and adolescents revealed lower mean scores across most subscales compared to younger children, particularly in the school setting. This could be indicative of a developmental trajectory where manifestations of ASD characteristics evolve or become less evident as individuals age. However, the clinical setting for this age group reflects a variance in the trend, particularly within the social communication and social interaction subscales, where mean scores are significantly higher, reinforcing the notion that clinical populations may present with more severe manifestations of ASD.

Adult participants, although limited in number and lacking a clinical comparison group, showed the highest mean scores in the emotional responses’ subscale within the school setting, potentially suggesting an accumulation of emotional expression challenges with age. The maladaptive speech subscale, however, did not follow this pattern, with mean scores similar to those of younger children.

The Autism Index, serving as a composite measure derived from the subscale scores, further underscores these patterns. Young children in clinical settings exhibit the highest mean scores, followed by adults in school settings, and then children and adolescents in clinical settings. This index, a critical component of the GARS-3, encapsulates the cumulative impact of ASD characteristics across subscales and substantiates the necessity of nuanced interpretation when comparing across age groups and settings.

These findings, situated within the confidence interval parameters, highlight the variability of ASD presentation. Such variability necessitates a discerning approach to ASD assessment, taking into account the age and setting of individuals, to ensure accurate and culturally sensitive diagnoses. The data underscore the complexity of ASD and the imperative of tailored assessments that consider the developmental and environmental contexts of the individuals assessed. Figure 4, Figure 5 and Figure 6 provide visualized graphs for these differences.

### 3.5. Inferential Statistical Analysis of Developmental Status, Geographical Location, Sex, Age, and Different Levels of ASD

In the third step, inferential statistics were employed to discern the differences among various covariates in relation to the Autism Index. Linear regression analysis was conducted (See Table 4), revealing a robust model with an R^2^ of 0.91883, indicating that approximately 91.88% of the variance in the Autism Index can be explained by the independent variables included in the model.

The omnibus ANOVA test within the linear regression analysis provided insight into the significance of each covariate. It was observed that age group (*p* = 0.886), presence or absence of psychiatric history (*p* = 0.529), sex (*p* = 0.769), and city (*p* = 0.193) were not significant predictors of the Autism Index scores. However, the subscales of restricted/repetitive behaviours (*p* < 0.001), social interaction (*p* = 0.002), social communication (*p* < 0.001), emotional responses (*p* = 0.001), cognitive style (*p* = 0.044), and maladaptive speech (*p* < 0.001) were found to be significant predictors, indicating that these domains of ASD-related characteristics bear a consequential impact on the overall assessment of ASD severity as measured by the Autism Index.

When examining the model coefficients, the influences of individual predictors on the Autism Index scores were further elucidated. Notably, the city of Taif demonstrated a significant positive relationship with the Autism Index (*p* = 0.024), suggesting that geographical location within Saudi Arabia may be associated with variability in ASD assessment outcomes. Moreover, the subscales of restricted/repetitive behaviours, social interaction, social communication, emotional responses, cognitive style, and maladaptive speech each significantly contributed to the prediction of the Autism Index scores, with standardized estimates reflecting their respective effect sizes.

This regression analysis underscores the multifaceted nature of ASD assessment through the A-GARS-3 and the complex interplay of various factors that contribute to the overall evaluation of ASD severity. The findings stress the importance of a comprehensive approach to ASD diagnosis, considering a broad spectrum of behaviours and communication styles, and highlight the potential influence of cultural and regional factors on the assessment outcomes.

In further examining the predictors of ASD severity within the Saudi Arabian context, the study progressed to a log-linear regression analysis (See Table 5). This statistical approach extended beyond the descriptive, delving into the multivariate relationships and interactions between several predictors including sex, age group, developmental status (typically or atypically developing), and the levels of ASD as categorized by DSM-5 severity (ranging from no ASD to very substantial support).

The model fit measures indicated a perfect fit (R^2^ = 1.00000), with the omnibus test reaching statistical significance (χ^2^ = 299.50, df = 47, *p* < 0.001). These results suggest that the model was highly effective in predicting the presence and severity of ASD based on the included predictors. However, it is noteworthy that the R^2^ value of 1.00000 may indicate overfitting, a scenario where the model too closely matches the sample data and may not generalize well to other datasets.

Within the model coefficients, the intercept was significant (*p* < 0.001), establishing a baseline relationship between the predictors and ASD severity. The main effects for age group and the setting did not reach statistical significance when compared to young children, nor did the main effect for sex. However, interactions between age group and psychiatric history, as well as between age group and DSM-5 severity levels, were notably significant. Particularly, the interaction between children and adolescents with no ASD compared to Level 1 severity (*p* = 0.003) and the interaction with Level 3 severity (*p* = 0.037) were significant, indicating that the association between age group and severity level of support needed is contingent upon the presence of a psychiatric history.

The coefficients for interactions involving the clinical setting were generally non-significant, with large standard errors, which suggests a degree of instability in the estimates likely due to small sample sizes in certain categories or potential data sparsity. This warrants a cautious interpretation of these interaction terms, as the results may not accurately represent the population.

In sum, the log-linear regression analysis provided a nuanced understanding of the multifactorial nature of ASD diagnosis and severity. It underscores the complexity of interactions between demographic variables and clinical presentations of ASD, and it highlights the necessity for comprehensive assessment protocols that account for the interplay of these variables. The results of this analysis are pivotal for informing clinical practices and tailoring intervention strategies to meet the individualized needs of the ASD population in Saudi Arabia.

## 4. Discussion

The primary aim of this study was to use the Arabic adaptation of the A-GARS-3 to investigate how ASD symptoms manifest across different demographic variables in Saudi Arabia, including developmental status, gender, age groups, and geographic regions. The study focused on evaluating ASD characteristics through six subscales: restricted/repetitive behaviors, social interaction, social communication, emotional responses, cognitive style, and maladaptive speech. Our analysis revealed notable differences in ASD presentations among the various demographic factors. We observed significant disparities in the likelihood and intensity of ASD symptoms in relation to setting, where clinical settings depicted a higher propensity and greater severity of ASD. This aligns with earlier research indicating that children with ASD often experience more severe symptoms associated with greater challenges [20,21,22]. Furthermore, age-related patterns suggest that younger children exhibit more severe ASD symptoms, while geographical analysis points to regional variances in the severity of ASD, corroborating with previous findings that younger age correlates with a higher level of ASD-related difficulties [23,24].

Interestingly, this study found that males were slightly more likely to be diagnosed with ASD compared to females across all settings, though some existing studies reported that sex differences of children with ASD are significantly limited and there are similarities and no differences [25,26]. There are similar studies that support the fact that males have been diagnosed with ASD more compared to female [27,28,29]. Similarly previous studies reported that females with ASD demonstrate less restricted and repetitive behaviors, although sex effects may be highlighted due to use of recognized, male-centric diagnostic instruments. Females also use better vocal expressiveness in how they use and alter the quality of their speech resulting in more natural speech patterns during social communication [30,31,32]. In the same vein, it was reported that ten years ago, a UK study found that, regardless of the severity of autistic traits, boys were more likely to receive a diagnosis than girls [33].

The study’s results highlight the complex interplay of demographic factors in the presentation of ASD, emphasizing the need for tailored assessment protocols. The findings of the research showed that the A-GARS-3 proved to be a reliable and culturally appropriate tool for evaluating ASD in the Saudi Arabian context. These findings suggest the necessity for age-specific and culturally informed diagnostic approaches in clinical and educational settings. The regional differences in ASD prevalence and severity call for targeted health policies and resource distribution to address the diverse needs across Saudi provinces. This finding indicated that social and demographic factors play a role in receiving a diagnosis of autistic spectrum disorder (ASD) separately of symptom severity and this supports the reality of socio-demographic influence found in the previous existing research [33,34].

Further, the previous validations of the GARS across different cultural contexts have consistently demonstrated the scale’s reliability and validity for assessing ASD when it is properly adapted and standardized. For example, the Spanish adaptation of GARS-2 exhibited high internal consistency and discriminative validity [35], a pattern also evident in the Turkish [36] and Korean versions [37,38]. These studies collectively affirm the universality of ASD and the cultural robustness of GARS across diverse populations [21,39,40].

In alignment with these international findings, our study has shown the Arabic version of the GARS-3 to be a reliable tool within the Saudi Arabian context. This is in agreement with previous research, such as the Jordanian translated Arabic version of GARS-2, which indicated significant validity and reliability indicators [41]. Our study’s focus on demographic variables, including developmental status, sex, age, and geographical region, echoes the recognized necessity for culturally sensitive approaches in the diagnosis of ASD. Although our sample size was smaller, the significant demographic differences observed in ASD manifestations are in line with the global understanding that GARS can be adapted to various cultures when proper translation and standardization are applied.

Moreover, the emphasis on the need for culturally sensitive tools, as highlighted by the Chinese validation studies [42], supports our findings that regional disparities in ASD presentation must be considered in diagnosis and assessment. Thus, our research both confirms the validity and reliability of the Arabic GARS-3 and emphasizes the critical role of cultural and demographic factors in assessing ASD. This supports the ongoing need for the adaptation and refinement of diagnostic instruments to meet the diverse needs of populations across different regions [39,40,43].

Additionally, Alasmari, Alduais, and Qasem’s study highlights the critical role of language in the assessment and diagnosis of ASD in Saudi Arabia, emphasizing the evolution of research towards a more culturally and linguistically informed approach [44]. This aligns with our findings underscoring the need for assessment tools that are sensitive to the linguistic nuances and cultural context of Saudi Arabian society, a perspective that complements our findings on the reliability of the Arabic GARS-3 for this population. This recent research points to a significant gap in longitudinal studies addressing language and cultural impacts on ASD, bolstering the call for tailored diagnostic practices in clinical settings within the region.

## 5. Limitations

The present study, while illuminating in its findings, is not without limitations that warrant circumspection. Primarily, its cross-sectional nature precludes the establishment of causality, rendering the temporal sequencing of ASD symptomatology and its predictors elusive. Additionally, the reliance on self-reported data via Google Forms, albeit innovative, may introduce self-selection bias, potentially skewing the sample towards individuals with internet access and technological proficiency. Furthermore, the psychometric properties, while robust, were evaluated within a homogeneous cultural context, which may limit the generalizability of the A-GARS-3 across Arab societies with varying dialects and cultural nuances. Finally, the sample size, particularly within the adult cohort, was relatively modest, constraining the statistical power to detect nuanced differences within this demographic, an issue compounded by the absence of a clinical comparison group for adults.

## 6. Implications for Practice

The findings of this study have substantial implications for the clinical assessment and intervention strategies of ASD within the Saudi Arabian context. The demonstrated reliability and validity of the A-GARS-3 affirm its utility as a culturally attuned diagnostic tool, facilitating earlier and more accurate identification of ASD, which is paramount for timely and effective intervention. The nuanced understanding of ASD presentation across different age demographics underscores the exigency for age-specific assessment protocols. Moreover, the regional disparities in ASD severity indices intimate the necessity for regionalized health policies and resource allocation to address the variegated needs across Saudi provinces. In the ambit of educational and clinical settings, the insights gleaned from this research advocate for tailored support programs that address both the common and idiosyncratic challenges faced by individuals with ASD.

## 7. Conclusions

In brief, this research utilized a cross-sectional design to assess how symptoms of ASD present within the Saudi Arabian context, with a focus on a sample of 178 participants to understand demographic differences. The diverse participant group allowed for a comprehensive analysis of the prevalence and severity of ASD across different ages, sexes, and developmental stages within the Saudi context. Innovative methods, such as employing Google Forms for data collection, facilitated an efficient and secure process, upholding the integrity of the data and preserving participant confidentiality. The findings provide nuanced insights into how demographic factors such as age, sex, and developmental status intersect with ASD presentations. These insights highlight the importance of a nuanced diagnostic approach that considers the complex interplay of individual characteristics with ASD. The study contributes valuable perspectives to the body of ASD research, offering guidance for future investigations into the nuanced expressions of autism spectrum disorders within specific cultural settings.

## Figures and Tables

**Figure 1 children-11-00472-f001:**
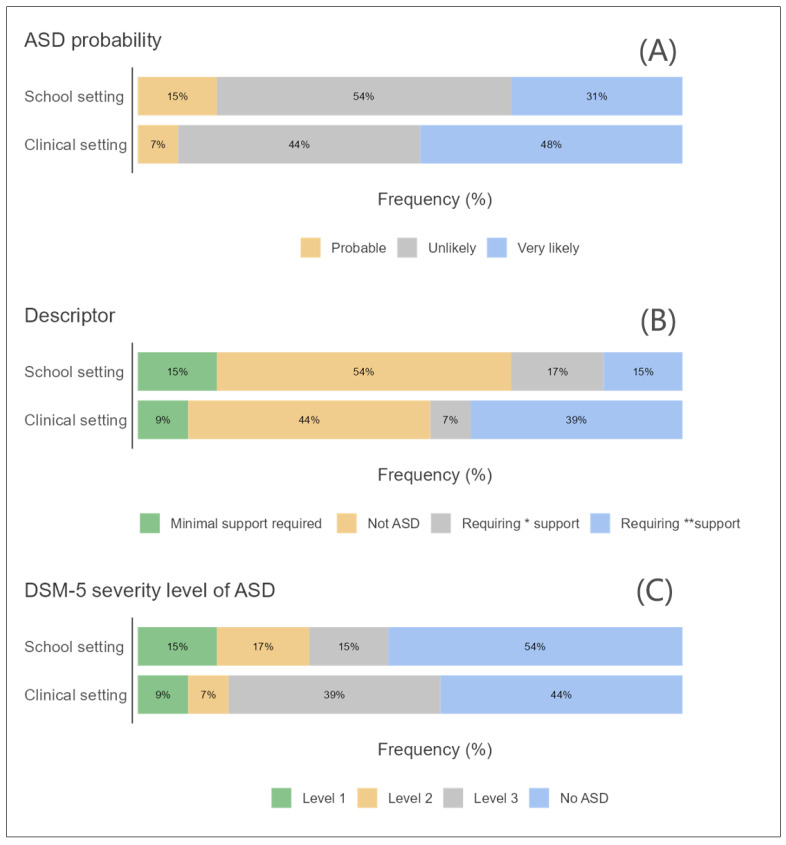
A survey plot for the distribution participants by (**A**) developmental status and ASD probability, (**B**) ASD description, and (**C**) DSM-5 severity level. * Substantial support. ** Very substantial support.

**Figure 2 children-11-00472-f002:**
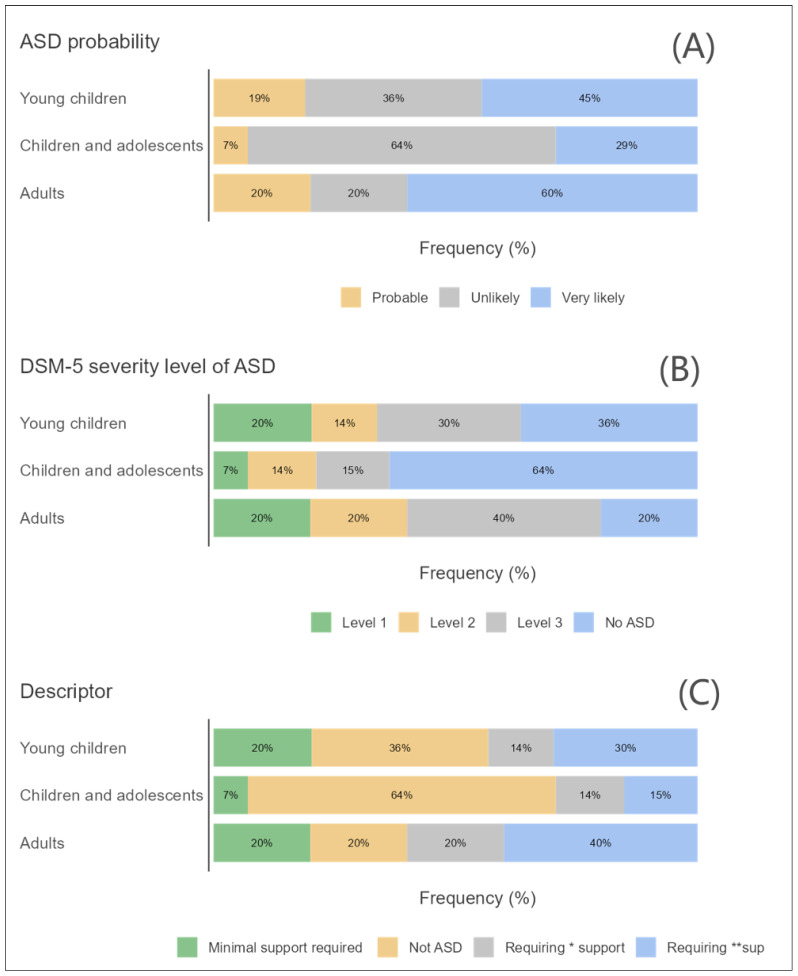
A survey plot for the distribution participants by (**A**) age group and ASD probability, (**B**) ASD description, and (**C**) DSM-5 severity level. * Substantial support. ** Very substantial support.

**Figure 3 children-11-00472-f003:**
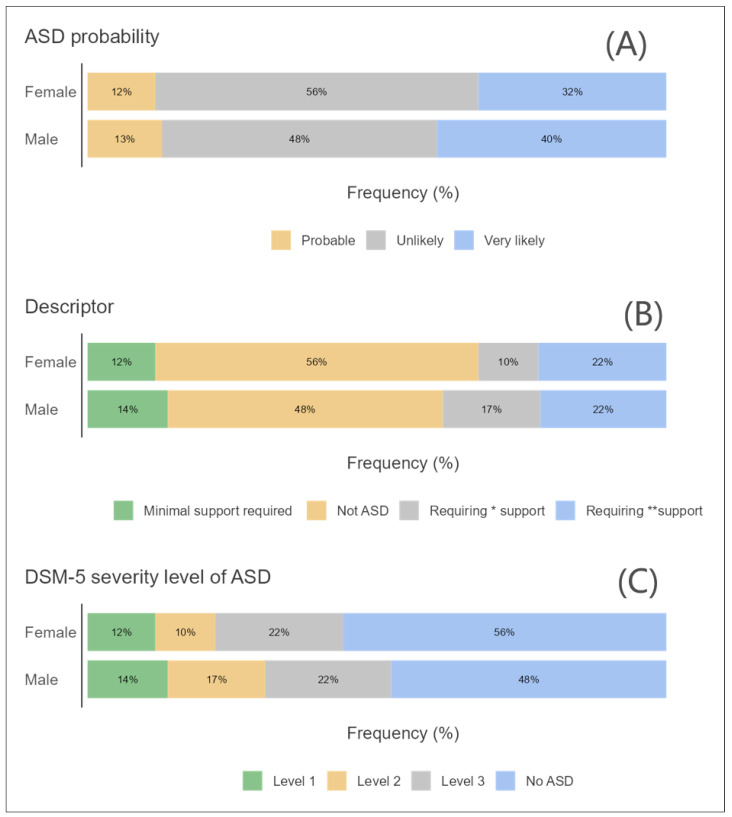
A survey plot for the distribution participants by (**A**) sex and ASD probability, (**B**) ASD description, and (**C**) DSM-5 severity level. * Substantial support. ** Very substantial support.

**Figure 4 children-11-00472-f004:**
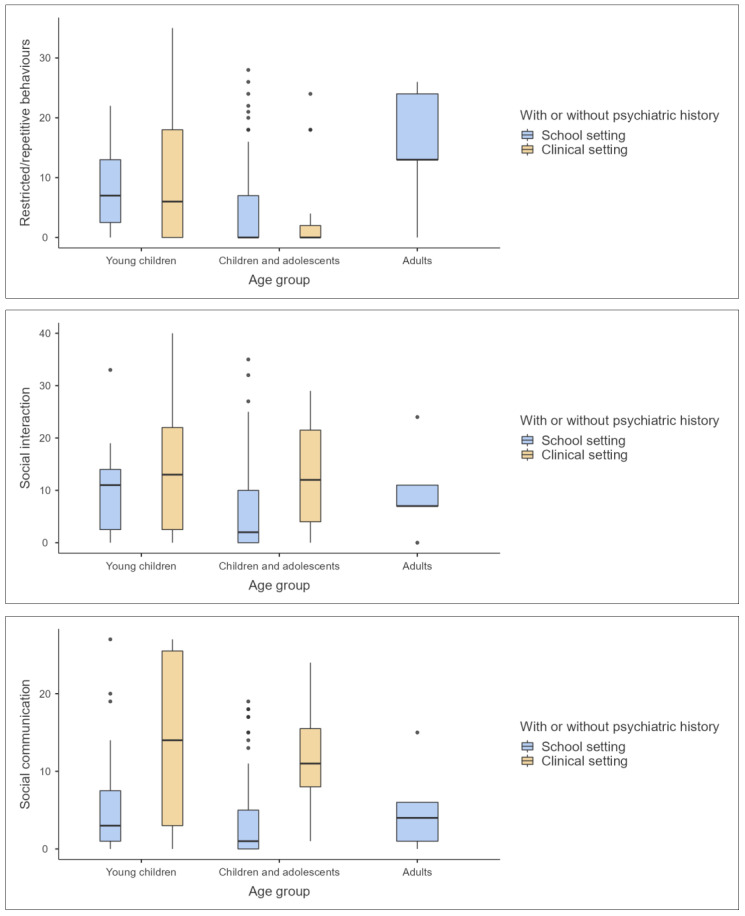
Graph plots for the first three subscales of A-GARS-3 by age group and developmental status.

**Figure 5 children-11-00472-f005:**
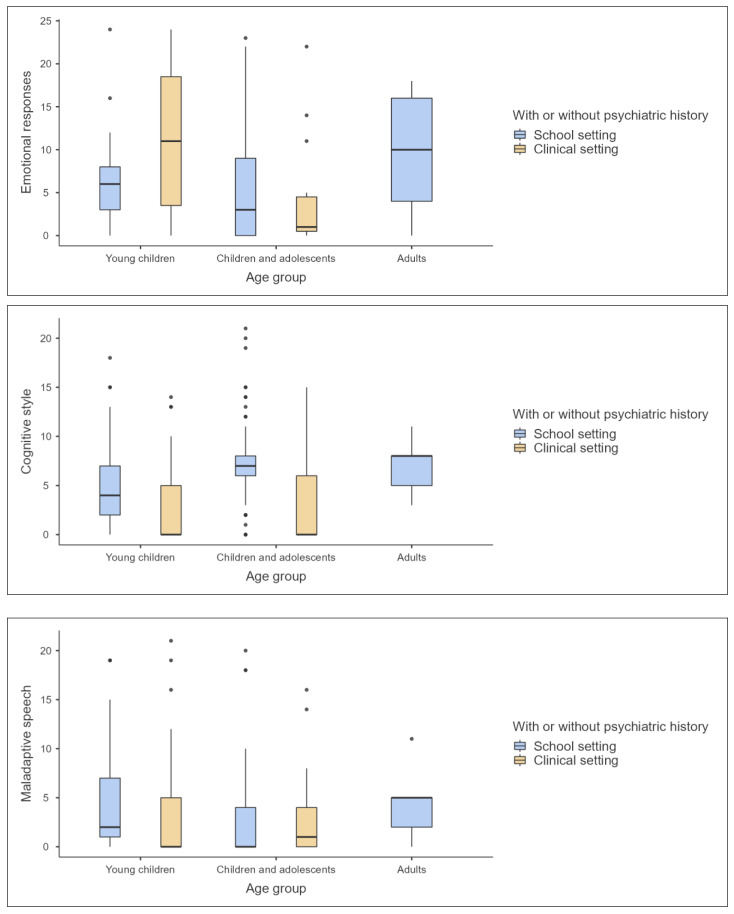
Graph plots for last three subscales of A-GARS-3 by age group and developmental status.

**Figure 6 children-11-00472-f006:**
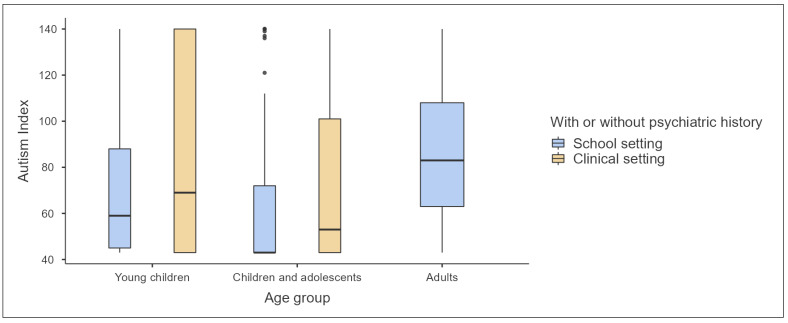
Autism index for A-GARS-3 by age group and developmental status.

**Table 1 children-11-00472-t001:** Demographic characteristics of participants.

	School Setting (n)	Clinical Setting (n)	%	
Age Group	124	54	69.66	30.34
3–5	28	28	22.58	51.85
6–10	59	21	47.58	38.89
11–15	28	5	22.58	9.26
16–20	6	0	4.84	0
21–22	3	0	2.42	0
**Gender Group**				
Female	60	17	48.39	31.48
Male	64	37	51.61	68.52
**City Group**				
Asir province	90	3	72.58	5.56
Riyadh Province	4	0	3.23	0
Eastern Province	10	0	8.06	0
Medina Province	1	0	0.81	0
Tabuk Province	0	1	0	1.85
Mecca Province	19	50	15.32	92.59
**Exceptionality Status**				
No exceptionality	124	NA	100	NA
Attention deficit hyperactivity disorder	NA	3	NA	5.56
Autism spectrum disorder	NA	16	NA	29.63
Apraxia of speech	NA	1	NA	1.85
Hearing impairment	NA	6	NA	11.11
Dysarthria	NA	1	NA	1.85
Speech and language delay	NA	27	NA	50.00
**Ethnicity**				
Arabs	124	54	69.66	30.34
**Language (mother tongue)**				
Arabic (Saudi dialect)	124	54	69.66	30.34

The sample included 178 participants with a mean age of 8.16 years (SD = 4.03). The participants were categorized into two groups: those in school settings (n = 124; M = 9.15, SD = 4.12) and those in clinical settings (n = 54; M = 5.89, SD = 2.70). The reported means and standard deviations reflect the participants’ ages in years.

**Table 2 children-11-00472-t002:** Subscales of the GARS-3 for measuring ASD.

Subscale	Elaboration on Relation to ASD Based on GARS-3
Restricted/Repetitive Behaviours	This subscale measures the presence and severity of behaviours characterized by rigidity, restricted interests, and repetitive actions, which are core features of ASD.
Social Interaction	Assesses the individual’s ability to engage in and maintain social reciprocity, recognize social cues, and form social connections, which are often impaired in ASD.
Social Communication	Evaluates the capacity for verbal and non-verbal communication, including the use of language for social engagement, often disrupted in individuals with ASD.
Emotional Responses	Probes the range, intensity, and appropriateness of emotional reactions, which can be markedly atypical or muted in persons with ASD.
Cognitive Style	Examines the individual’s thinking patterns, problem-solving approach, and preference for detail, which may be rigid or idiosyncratic in ASD.
Maladaptive Speech	Measures aspects of speech and language such as echolalia, pronoun reversal, or atypical language processing, commonly observed in ASD.

**Table 3 children-11-00472-t003:** Descriptive analysis of ASD by age group and developmental status across the six subscales of A-GARS-3.

					95% Confidence Interval				
ASD Covariates	Age Group	Setting	*n*	Mean	Lower	Upper	SD	Minimum	Maximum
Restricted/repetitive behaviours	Young children	School	35	8.3143	5.910032	10.7185	6.9990	0	22
		Clinical	39	9.8974	6.373936	13.4209	10.8695	0	35
	Children and adolescents	School	84	4.4048	2.875758	5.9338	7.0457	0	28
		Clinical	15	4.2667	−0.335610	8.8689	8.3106	0	24
	Adults	School	5	15.2000	2.254499	28.1455	10.4259	0	26
		Clinical	0	NaN	NaN	NaN	NaN	NaN	NaN
Social interaction	Young children	School	35	9.3429	6.871448	11.8143	7.1945	0	33
		Clinical	39	13.7436	9.718284	17.7689	12.4176	0	40
	Children and adolescents	School	84	6.2262	4.443530	8.0089	8.2145	0	35
		Clinical	15	12.7333	7.362318	18.1043	9.6988	0	29
	Adults	School	5	9.8000	−1.215176	20.8152	8.8713	0	24
		Clinical	0	NaN	NaN	NaN	NaN	NaN	NaN
Social communication	Young children	School	35	5.3143	3.057461	7.5711	6.5699	0	27
		Clinical	39	13.7949	10.342419	17.2473	10.6504	0	27
	Children and adolescents	School	84	3.5000	2.362845	4.6372	5.2400	0	19
		Clinical	15	11.3333	7.941051	14.7256	6.1257	1	24
	Adults	School	5	5.2000	−2.218877	12.6189	5.9749	0	15
		Clinical	0	NaN	NaN	NaN	NaN	NaN	NaN
Emotional responses	Young children	School	35	6.2000	4.464749	7.9353	5.0515	0	24
		Clinical	39	10.6667	8.046492	13.2868	8.0829	0	24
	Children and adolescents	School	84	4.8333	3.580180	6.0865	5.7745	0	23
		Clinical	15	4.2667	0.699383	7.8340	6.4417	0	22
	Adults	School	5	9.6000	0.078777	19.1212	7.6681	0	18
		Clinical	0	NaN	NaN	NaN	NaN	NaN	NaN
Cognitive style	Young children	School	35	5.2286	3.572561	6.8846	4.8208	0	18
		Clinical	39	2.8974	1.518527	4.2763	4.2538	0	14
	Children and adolescents	School	84	7.2857	6.404317	8.1671	4.0615	0	21
		Clinical	15	3.2667	0.848558	5.6848	4.3665	0	15
	Adults	School	5	7.0000	3.172935	10.8271	3.0822	3	11
		Clinical	0	NaN	NaN	NaN	NaN	NaN	NaN
Maladaptive speech	Young children	School	35	4.6857	2.891520	6.4799	5.2231	0	19
		Clinical	39	3.4872	1.682516	5.2918	5.5672	0	21
	Children and adolescents	School	84	2.4286	1.478571	3.3786	4.3776	0	20
		Clinical	15	3.4000	0.493683	6.3063	5.2481	0	16
	Adults	School	5	4.6000	−0.564486	9.7645	4.1593	0	11
		Clinical	0	NaN	NaN	NaN	NaN	NaN	NaN
Autism Index	Young children	School	35	71.6000	61.495909	81.7041	29.4141	43	140
		Clinical	39	89.9487	75.363880	104.5336	44.9924	43	140
	Children and adolescents	School	84	61.2738	54.628896	67.9187	30.6198	43	140
		Clinical	15	75.0000	55.051782	94.9482	36.0218	43	140
	Adults	School	5	87.4000	40.211867	134.5881	38.0039	43	140
		Clinical	0	NaN	NaN	NaN	NaN	NaN	NaN

Note. The CI of the mean assumes sample means follow a t-distribution with N − 1 degrees of freedom. NaN: Not a Number, calculations were not figured out statistically.

**Table 4 children-11-00472-t004:** Model coefficients—Autism Index and GARS-3 subscales by age group, developmental status, sex, and city.

Predictor	Estimate	SE	*t*	*p*	Stand. Estimate
Intercept ^a^	36.25922	3.87145	9.365798	<0.001	
Age group:					
Children and adolescents − Young children	0.25656	2.18107	0.117629	0.907	0.0070924
Adults − Young children	3.51402	7.16343	0.490550	0.624	0.0971435
With or without psychiatric history:					
Clinical setting − School setting	3.70333	5.87191	0.630686	0.529	0.1023769
Sex:					
Male − Female	−0.59044	2.00372	−0.294673	0.769	−0.0163225
City:					
Alahsa − Abha	2.89860	7.34399	0.394690	0.694	0.0801304
Aldammam − Abha	−9.78531	6.45331	−1.516324	0.132	−0.2705103
Alnamas − Abha	−0.26859	3.36240	−0.079880	0.936	−0.0074250
Bisha − Abha	−5.59268	3.80554	−1.469615	0.144	−0.1546071
Damam − Abha	9.73980	8.80952	1.105600	0.271	0.2692524
Jedah − Abha	7.73624	8.46106	0.914335	0.362	0.2138648
Jeddah − Abha	−5.14428	6.79038	−0.757583	0.450	−0.1422112
Madina − Abha	−0.22739	11.68801	−0.019455	0.985	−0.0062860
Meca − Abha	10.56277	6.67077	1.583440	0.115	0.2920029
Mecca − Abha	0.15357	8.23633	0.018646	0.985	0.0042454
Riyadh − Abha	1.77468	11.70869	0.151570	0.880	0.0490602
Riyadh − Abha	3.74127	8.78265	0.425985	0.671	0.1034258
Riyadh − Abha	11.06206	11.92286	0.927802	0.355	0.3058056
Tabuk − Abha	14.58953	12.84005	1.136252	0.258	0.4033210
Taif − Abha	28.07248	12.31899	2.278797	0.024	0.7760508
Jeddah − Abha	−4.96632	7.07335	−0.702116	0.484	−0.1372915
Restricted/repetitive behaviours	1.23866	0.18274	6.778242	<0.001	0.2941276
Social interaction	0.57525	0.18132	3.172541	0.002	0.1539529
Social communication	1.53008	0.20861	7.334672	<0.001	0.3474993
Emotional responses	0.72660	0.21869	3.322467	0.001	0.1347893
Cognitive style	0.54846	0.26989	2.032135	0.044	0.0699885
Maladaptive speech	1.29200	0.24114	5.357834	<0.001	0.1758611

^a^ Represents reference level. Note: SE (Standard Error): Measures the precision of a sample mean, indicating how much it can vary. *t* (*t*-Statistic): Used in *t*-tests, it represents the standardized difference between a sample statistic and a hypothesized value. *p* (*p*-Value): Provides the probability of observing the data, or more extreme, if the null hypothesis is true, used for hypothesis testing.

**Table 5 children-11-00472-t005:** Log linear regression: model coefficients by age group, developmental status, DSM-5 severity level, and sex.

Predictor	Estimate	SE	Z	*p*
Intercept	1.791759	0.40825	4.388896	<0.001
Age group:				
Children and adolescents − Young children	−1.098612	0.81650	−1.345520	0.178
Adults − Young children	−27.094345	189,338.66108	−1.4310 × 10^−4^	1.000
With or without psychiatric history:				
Clinical setting − School setting	−1.791759	1.08012	−1.658847	0.097
DSM-5 severity level of ASD:				
Level 2 − Level 1	−0.405465	0.64550	−0.628144	0.530
Level 3 − Level 1	−0.693147	0.70711	−0.980258	0.327
No ASD − Level 1	−0.182322	0.60553	−0.301094	0.763
Sex:				
Male − Female	−0.405465	0.64550	−0.628144	0.530
Age group * With or without psychiatric history:				
(Children and adolescents − Young children) * (Clinical setting − School setting)	−24.203973	189,338.66083	−1.2783 × 10^−4^	1.000
(Adults − Young children) * (Clinical setting − School setting)	1.791759	267,765.30214	6.6915 × 10^−6^	1.000
Age group * DSM-5 severity level of ASD:				
(Children and adolescents − Young children) * (Level 2 − Level 1)	0.405465	1.19024	0.340659	0.733
(Adults − Young children) * (Level 2 − Level 1)	0.405465	267,765.30227	1.5143 × 10^−6^	1.000
(Children and adolescents − Young children) * (Level 3 − Level 1)	2.197225	1.05409	2.084470	0.037
(Adults − Young children) * (Level 3 − Level 1)	0.693147	267,765.30213	2.5886 × 10^−6^	1.000
(Children and adolescents − Young children) * (No ASD − Level 1)	2.821379	0.94994	2.970068	0.003
(Adults − Young children) * (No ASD − Level 1)	25.484907	189,338.66108	1.3460 × 10^−4^	1.000
With or without psychiatric history * DSM-5 severity level of ASD:				
(Clinical setting − School setting) * (Level 2 − Level 1)	−24.897120	189,338.66068	−1.3150 × 10^−4^	1.000
(Clinical setting − School setting) * (Level 3 − Level 1)	2.079442	1.32288	1.571910	0.116
(Clinical setting − School setting) * (No ASD − Level 1)	1.974081	1.23828	1.594214	0.111
Age group * Sex:				
(Children and adolescents − Young children) * (Male − Female)	1.321756	1.05672	1.250805	0.211
(Adults − Young children) * (Male − Female)	25.708050	189,338.66108	1.3578 × 10^−4^	1.000
With or without psychiatric history * Sex:				
(Clinical setting − School setting) * (Male − Female)	1.791759	1.29099	1.387891	0.165
DSM-5 severity level of ASD * Sex:				
(Level 2 − Level 1) * (Male − Female)	0.628609	0.93095	0.675234	0.500
(Level 3 − Level 1) * (Male − Female)	−4.4186 × 10^−15^	1.11803	−3.9521 × 10^−15^	1.000
(No ASD − Level 1) * (Male − Female)	0.587787	0.88506	0.664120	0.507
Age group * With or without psychiatric history * DSM-5 severity level of ASD:				
(Children and adolescents − Young children) * (Clinical setting − School setting) * (Level 2 − Level 1)	50.892852	267,765.30190	1.9007 × 10^−4^	1.000
(Adults − Young children) * (Clinical setting − School setting) * (Level 2 − Level 1)	24.897120	423,374.11640	5.8806 × 10^−5^	1.000
(Children and adolescents − Young children) * (Clinical setting − School setting) * (Level 3 − Level 1)	21.719066	189,338.66083	1.1471 × 10^−4^	1.000
(Adults − Young children) * (Clinical setting − School setting) * (Level 3 − Level 1)	−2.079442	378,677.32166	−5.4913 × 10^−6^	1.000
(Children and adolescents − Young children) * (Clinical setting − School setting) * (No ASD − Level 1)	21.788059	189,338.66083	1.1507 × 10^−4^	1.000
(Adults − Young children) * (Clinical setting − School setting) * (No ASD − Level 1)	−27.276666	327,944.18045	−8.3175 × 10^−5^	1.000
Age group * With or without psychiatric history * Sex:				
(Children and adolescents − Young children) * (Clinical setting − School setting) * (Male − Female)	−2.708050	267,765.30179	−1.0114 × 10^−5^	1.000
(Adults − Young children) * (Clinical setting − School setting) * (Male − Female)	−27.094345	327,944.18044	−8.2619 × 10^−5^	1.000
Age group * DSM-5 severity level of ASD * Sex:				
(Children and adolescents − Young children) ✻ (Level 2 − Level 1) * (Male − Female)	−0.040822	1.47573	−0.027662	0.978
(Adults − Young children) * (Level 2 − Level 1) * (Male − Female)	−0.628609	267,765.30227	−2.3476 × 10^−6^	1.000
(Children and adolescents − Young children) * (Level 3 − Level 1) * (Male − Female)	−2.420368	1.60035	−1.512402	0.130
(Adults − Young children) * (Level 3 − Level 1) * (Male − Female)	0.693147	267,765.30214	2.5886 × 10^−6^	1.000
(Children and adolescents − Young children) * (No ASD − Level 1) * (Male − Female)	−1.540445	1.24743	−1.234894	0.217
(Adults − Young children) * (No ASD − Level 1) * (Male − Female)	−51.192957	267,765.30216	−1.9119 × 10^−4^	1.000
With or without psychiatric history * DSM-5 severity level of ASD * Sex:				
(Clinical setting − School setting) * (Level 2 − Level 1) * (Male − Female)	23.287682	189,338.66069	1.2299 × 10^−4^	1.000
(Clinical setting − School setting) * (Level 3 − Level 1) * (Male − Female)	−0.207639	1.68135	−0.123496	0.902
(Clinical setting − School setting) * (No ASD − Level 1) * (Male − Female)	−1.463255	1.51658	−0.964842	0.335
Age group * With or without psychiatric history * DSM-5 severity level of ASD * Sex:				
(Children and adolescents − Young children) * (Clinical setting − School setting) * (Level 2 − Level 1) * (Male − Female)	−24.568616	327,944.18010	−7.4917 × 10^−5^	1.000
(Adults − Young children) * (Clinical setting − School setting) * (Level 2 − Level 1) * (Male − Female)	−23.287682	500,943.01009	−4.6488 × 10^−5^	1.000
(Children and adolescents − Young children) * (Clinical setting − School setting) * (Level 3 − Level 1) * (Male − Female)	3.726620	267,765.30179	1.3917 × 10^−5^	1.000
(Adults − Young children) * (Clinical setting − School setting) * (Level 3 − Level 1) * (Male − Female)	−0.485508	463,783.10756	−1.0468 × 10^−6^	1.000
(Children and adolescents − Young children) * (Clinical setting − School setting) * (No ASD − Level 1) * (Male − Female)	2.926739	267,765.30179	1.0930 × 10^−5^	1.000
(Adults − Young children) * (Clinical setting − School setting) * (No ASD − Level 1) * (Male − Female)	52.068426	463,783.10754	1.1227 × 10^−4^	1.000

Note: * The asterisk (*) symbol is used to denote an interaction between two categorical variables. This means that the analysis is not only examining the individual effects of each variable on the outcome but also how these variables work together to influence the outcome.

## Data Availability

The raw data supporting the conclusions of this article will be made available by the corresponding author on request. The data are not publicly available due to confidentiality and privacy disclosure of the participants’ identities and authorized sharing of the data.

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
