# Peer review of "Examining Language, Speech and Behaviour Characteristics: A Cross-Sectional Study in Saudi Arabia Using the Arabic Version of Gilliam Autism Rating Scale-Third Edition"

_children, 2024, doi:10.3390/children11040472_

Round 1
Reviewer 1 Report
Comments and Suggestions for Authors
Establishing the prevalence and characteristics of autism in countries such as Saudi Arabia is a valid endeavor that will support considerations for intervention. Thus, this study has the potential to be quite useful. However, there are many challenges in this manuscript, delineated below, that make the value of outcomes uncertain.
Introduction
· Is there a citation for the quote from Dr Shore?
· The entire section 1.1 is in need of citations.
· Overall, the introduction seems disconnected from the aim of the study. There is no background presented that addresses developmental status, sex, or geographical locations anywhere, let alone Saudi Arabia.
· Why study Saudi Arabia in particular? Why consider that autism might be differently expressed in this country than it is in the USA? There is of course inherent logic in making this assumption, but authors need to provide some background on the cultural considerations that drove this study.
Methods
· Provide evidence of ethical approval; please indicate how ethical approval was obtained.
· Describe the recruitment process for children with autism and typically developing children, and the random sampling process. Were the participants with autism previously identified as being autistic? A more detailed description of the educational and clinical samples is needed. An explanation of what constitutes “exceptionality” is also needed.
· Citations are needed for the GARS-3 and the noted Arabic adaptation. The adaptation process is nicely described, but no citation is provided.
· Citations are needed to support the statements about the psychometrics of the GARS-3 and of the Arabic adaptation. Establishment of this adaptation as valid and reliable would have been better suited to a separate manuscript.
· The design description, section 2.3, is very complex and difficult to follow. State the study design in straight forward terms and revise this paragraph to more clearly present design elements.
· It is very unclear how study design might ‘elucidate’ the psychometric properties of the A-GARS-3; please clarify.
· The study aim stated in this section is to authenticate the A-GARS-3, but this differs from the aim stated at the end of the Introduction. There is a need to clearly and consistently state the study design.
· There is a need to more thoroughly describe and validate the training process used for the A-GARS-3 to train parents and teachers.
Results
· It is unclear why the likelihood of ASD was a factor given the diagnostic process; a description of the diagnostic process is needed. With a more thorough description of the sample in Methods (as requested above), this issue may be cleared up.
· As described, the initial phase of analysis does not match the purpose of this study. There should be consistency between aims throughout this manuscript.
· Analyses need to report the n along with standard deviation or some measure of variability.
· Figures require more clear labels for subsections (1A, 1B, etc).
· Required support: is this calculated for the entire sample or only those children identified as autistic?
· Figure 2A suggests that ASD probability is highest in adults; this differs from what is stated in the text.
· Figure 2B does not obviously include only children from the school setting.
· Overall, the figures 1-3 do not match well with the narrative, making it difficult to understand findings.
· The descriptive analysis does not present a statistical analysis of differences across groups of any sort, yet the narrative suggests otherwise. This is problematic and must be addressed.
· The inclusion of “psychiatric history” in the linear regression is surprising. How was this determined? What is the overall incidence? How could it be identified in young children? Answers to these questions may have influenced the findings.
Discussion
· The main purpose stated at the beginning of this section differs from that initially presented.
· Developmental status is not the same as setting, yet they appear to be equated in line 401.
· Clarify what is meant by “…ASD who usually suffer from severity autism symptoms that correlated with the severity problems…”
· Greater clarity is needed in the Results section to establish the noted age-related trends.
· Comments regarding female autistics are interesting but not examined at all in this study.
· Conclusions regarding the A-GARS-3 are outside of the main intent of this study.
Comments on the Quality of English LanguageEnglish/grammar: please edit for errors in both, there are many. As it stands these errors make understanding this manuscript challenging.
Author Response
"Please see the attachment."

Reviewer 2 Report
Comments and Suggestions for Authors
Thank you very much for giving me the opportunity to review this interesting manuscript titled “Examining Autism Spectrum Disorder Characteristics in Saudi Arabia: A Cross-sectional Study Using the Arabic Version of Gilliam Autism Rating Scale-Third Edition”.
With some improvements, the manuscript has the potential to be useful in the field. Overall the manuscript is appropriate for the scope of the Journal. The writing style is appropriate for a scientific manuscript. Although there are a few suggestions:
Introduction
- In lines 40-43 the authors mentioned the impact only in individuals with ASD. It would be helpful to the readers, some extra information to be added about the impact on the family quality of life etc. Reference to be mentioned: https://www.ncbi.nlm.nih.gov/pmc/articles/PMC10228246/
- Please add information about the diagnostic criteria – ICD and DSM -V.
- The introduction the paper’s aim. The aim of the study is general and the sample was small for this scope (only 178 participants). The authors should carefully rewrite the aim of the manuscript in more reasonable manner.
Methods:
- In lines 139 -145: The authors should add information about the Cronbach a in English version. Also, the English version reference should be mentioned.
- Lines 146-56: This paragraph should be titled as translation process with more information.
- The authors mentioned that the instrument had shown reliability and validity and a high internal consistency. However there no values, tables about these essential results in this manuscript or any further reference. Please, revise this section with the appropriate results that ensures the aforementioned.
Results
Well presented section with detail and useful tables. Please, revise this section according the suggestion above.
Discussion:
- Add references: When discussing that the main purpose of the study is to utilize A-GARS-3 adapted for Arabic speakers in Saudi Arabia, you need to enhance this section with a paragraph with other studies that adapted A-GARS-3 in other languages to compare and to discuss further the results.
- Moreover, the authors should revise this section according with the results of the validation of the instrument.
- Although limitations are discussed at the conclusion of this section, it is often considered preferable to discuss limitations closer to the presentation of the findings. This helps comprehend the surrounding circumstances and possible biases present in the research outcomes.
Author Response
"Please see the attachment."

Reviewer 3 Report
Comments and Suggestions for Authors
Examining Autism Spectrum Disorder Characteristic in Saudi Arabia: A Cross-sectional Study Using the Arabic Version of Gilliam Autism Rating Scale-Third Edition
Review
A brief summary
The study indicates that demographic factors can influence the assessment and diagnosis of Autism Spectrum Disorder (ASD).
General concept comments
Article:
v The paper is well written and organized.
Review:
v In order to make the text clearer ,Table 1 should be represented as histogram for : Age group , Gender group and City Group.
v Figures should be in better resolution.
v Titles in Tables 4 and 5 should be further explained : "SE t p " " SE Z p"
Specific comments:
Reference 17 about "GARS-3" should be mentioned at the first time where "GARS-3" appears i.e. line 132.
Comments on the Quality of English LanguageAuthor Response
"Please see the attachment."

Round 2
Reviewer 1 Report
Comments and Suggestions for Authors
As noted previously, the broad intent of this work is laudable and much needed. Thus, a study such as this study has the potential to be quite useful. However, there continue to be many challenges in this manuscript, delineated below, that make the value of outcomes uncertain. Authors were not entirely responsive to questions posed in the last review.
Introduction
· Autism is presented strictly as a disorder in this manuscript, without consideration given to the growing perspective that autism be perceived as a difference in neural function rather than a disorder. Authors should at the least acknowledge this different perspective, even though in this manuscript they reject it.
· The detailed diagnostic description that opens the paper, including the comparison between the ICD-11 criteria and DSM-5-TR criteria, is not needed. What was requested was background on the factors that this research examined as part of the aim.
· The information on family quality of life (lines 47-54) is partially repeated in the next paragraph (lines 64-71). Both the difference in font color and the similarities in content make it seem that these additions were created by different contributors. The manuscript should read like it comes from a single voice, and is not repetitive.
· Structurally the manuscript now jumps from section 1. to section 1.3, omitting 1.1 and 1.2.
· The final paragraph repeats to some extent what has also been previously stated.
Methods
· Methods start with the aim of assessing the utility of an Arabic adaptation of the GARS-3. This is not what is stated in the objectives presented in section 1.3.
· Provide evidence of ethical approval at the start of the methods section. This should come before the description of the sample. Simply stating that authors endeavored to insure ethical procedures were followed is insufficient.
· In describing the sample, please begin with recruitment. There is not a description of this process.
· Please define “exceptionality status” in the narrative.
· Citations continue to be needed for the GARS-3 the first time it is introduced and the noted Arabic adaptation also when first introduced. Is this now a published tool? Is it fully represented in citation #18 such that someone else could use it? It is not appropriate to cite the original GARS-3 to support the translation process for the A-GARS-3 (line 184). Please indicate in the text that the study documenting the psychometric properties of the A-GARS-3 is under review.
· Study design and procedures, sections 2.3 and 2.4, would be better placed at the beginning of methods, before participants. The statement regarding ethical approval should be accompanied by the approval number and moved to the beginning of the Methods section.
Results
· The first set of graphs, 1A-C looks at differences across settings (clinical vs school); this is not part of either study aim. This approach is apparent in other analyses as well, looking at setting. This makes results confusing throughout.
· Authors now indicate that these data were drawn from the GARS-2, and not the Arabic version. How valid is this data? When and for what analyses was the A-GARS-3 used? The use of the GARS-2 data needs to be explained in the Methods section.
· The intent to include both groups of children in Figure 2, as explained in the Response Letter is interesting, but actually addresses a research question authors have not posed. As such, the rationale for including both groups of children causes confusion.
· Figure 2A does not indicate that autism is more likely to be diagnosed in adults; this figure shows that the probability is nearly identical between young children and adults.
· Figure 2B does not clearly include only children from the school setting as previously stated. If authors are making the point that combining the minimal support and not ASD groups is evidence that this is only a school setting sample, this requires explanation in the text. A figure legend would help.
· Figure 2B: are the children identified in the “minimal support” category identified as having or not having autism? This figure remains unclear. A figure legend would help.
· Figure 3 has similar problems; what is presented is not clear.
· All figures need to include n
· Mean values shown in Table 3 fall within the range of the CI for young children for restricted/repetitive behaviors. Thus this is not statistically significant.
· Line 347: “Children and adolescents revealed lower means” Lower than what?
· Line 355: “adults showed the highest mean scores in emotional response subscale”, highest relative to their response on other subscales or relative to the response of other age groups?
· If the Autism Index is important, present this data.
· The inclusion of “psychiatric history” in the linear regression is surprising and, in the text, remains unclear. Authors offered an explanation to the review but did not explain the inclusion of this variable in the article.
Discussion
· The main purpose stated at the beginning of this section differs from that initially presented in that it does not include mention of autism severity.
· If the intent was to use the A-GARS-3, why are data presented from the GARS-2?
· It is unclear what data reflect the GARS-2 and what data reflect the A-GARS-3. As such it is challenging to determine the validity of conclusions.
Comments on the Quality of English LanguageAttention was paid to revising English in this version.
Author Response
"Please see the attachment."

Reviewer 2 Report
Comments and Suggestions for Authors
Good effort! The authors revised the manuscript, which is acceptable for publication.
Author Response
"Please see the attachment."
